# Real-Time Laser Tracker Compensation of Robotic Drilling and Machining

**Zheng Wang \***[ID]**, Runan Zhang and Patrick Keogh**

Department of Mechanical Engineering, University of Bath, Bath BA2 7AY, UK; rz216@bath.ac.uk (R.Z.); enspsk@bath.ac.uk (P.K.)

**\*** Correspondence: zw215@bath.ac.uk

**Abstract:** Due to their flexibility, low cost and large working volume, 6-axis articulated industrial robots are increasingly being used for drilling, trimming and machining operations, especially in aerospace manufacturing. However, producing high quality components has demonstrated to be difficult, as a result of the inherent problems of robots, including low structural stiffness and low positional accuracy. These limit robotic machining to non-critical components and parts with low accuracy and surface finish requirements. Studies have been carried out to improve robotic machine capability, specifically positioning accuracy and vibration reduction. This study includes the description of the hardware, software and methodologies developed to compensate robot path errors in real time using a single three-degrees-of-freedom (DOF) laser tracker, as well as the experimental results with and without compensation. Performance tests conducted include ballbar dynamic path accuracy test, a series of drilling case studies and a machining test. The results demonstrate major improvements in path accuracy, hole position accuracy and hole quality, as well as increases in accuracy of a machined aluminum part.

**Keywords:** industrial robot; laser tracker; real time; compensation

---

## 1. Introduction

At a fundamental kinematic level, an industrial robot has the same path motion potential as a traditional 5- or 6-axis machine tool, while offering lower cost, larger working volume and higher flexibility. Therefore, understandably, there is a large amount of commercial interest in utilizing the technology, especially in the machining, drilling or fettling of large components in aerospace manufacturing.

However, the reality of robotic machining leaves much to be desired. Robots have low positional accuracy, often worse than 0.4 mm, compared to 0.01 mm of a typical machine tool, as well as low stiffness, often 1—2 orders of magnitude below that of a machine tool [1–4]. This limits the robotic machining to soft materials such as foam and plastic and other niche applications that have low accuracy or low material removal rate requirements [5].

The current state-of-the-art for improving robot accuracy includes offline calibration and modeling of robot kinematics and online compensation of robot path using external instruments.

Although offline calibration can significantly improve the static accuracy of robots [6–8], when applied to "real world" problems they often prove to be inadequate [9,10]. This arises due to the parameters that are difficult to calibrate (e.g., thermal expansion, backlash, dynamic changes in load), which become dominant sources of error. In order to overcome the limitations of the offline calibration, it becomes necessary to directly correct the robot motion using metrology instruments in real time [11,12]. Typically, this involves target(s) on the end-effector tracked by metrology instruments, a controller/PC that generates the correction commands, which communicates with the robot controller.



The results are usually better than offline calibration, but the system required is much more complex and expensive [13,14].

Typical closed loop online compensation setups require very expensive 6-degrees-of-freedom (6-DOF) capable laser trackers or high-speed photogrammetry systems, putting them out of reach of most potential users.

To achieve even higher dynamics than the robot/controller can provide, high-dynamic end-effectors have also been developed. These are relatively small motion stages that have faster response times and can work independently from the robot controller [15,16]. However, these systems are currently restricted to having the spindle being stationary while the robot holds the part due to their size and complexity.

Since setting up online compensation on an industrial robot is an expensive and complex task, there is limited information on how such systems perform under more realistic conditions. While many papers have been published on machining with industrial robots, very few have attempted to correct their position errors in real-time using external metrology instruments. Of the papers focused on real-time metrology compensation, all used more expensive full six DOF metrology solutions (Leica trackers or Nikon photogrammetry [9,10] or directed laser photogrammetry using 4× modified FARO trackers [17]). Moreover, even fewer papers presented representative machining results (i.e., aluminum, or other metallic components, complex toolpath). Furthermore, many existing papers also do not describe fully the calibration process of the end-effector, which can have an impact on robot accuracy.

This study presents the process of setup, calibration of the end-effector, coordinate transformations and the control system required for real-time compensation of a standard KUKA industrial robot. The experimental results including dynamic path accuracy, drilling and machining trials using a simple spindle end-effector with real-time position compensation from a single FARO 3-DOF laser tracker are presented. Although a 3-DOF laser tracker cannot correct end-effector rotations errors, it can produce enough improvements to the robot performance to be worth the investment, considering the rather large cost difference between 3- and 6-DOF capable trackers.

Apart from the experimental results, several new methods are also included in this study. First, we present our new method of path corrections which significantly reduces the effect of jitter due to desync between the robot and the laser tracker. This allows other slower metrology instruments to be used. Second, we fully describe our simple and repeatable SMR calibration process. Third, our proposed method to mitigate backlash problems are greatly improved compared to previous work [18] where 10%–15% reduction was achieved.

Compared to previous work on industrial robots of similar performance by Droll et al. and Moeller et al. [13,14], we achieved better results of below 0.04 mm vs. 0.1–0.2 mm for similar tool path (slotting periphery, Section 8). The previous authors machined test pieces from a soft material when 3/6 DOF laser compensation is enabled, with simple slot geometries. More complex tool path in aluminum were trialed in our experiments, more representative of realistic machining operations. Schneider et al. experimented machining of steel with a KUKA KR125 compensated with a Nikon system, achieving 0.063 mm mean error for a circular path [10], compared to 0.011 mm (periphery, Figure 30) with our setup using the same metric. With our study, we also aim to help other interested parties in creating their own real-time compensation setups by sharing our methods, results and potential difficulties.

## 2. The LIMA Robot Cell

The experiments were carried out on the Laboratory for Integrated Metrology Applications (LIMA) robot cell at the University of Bath (Figure 1), on a standard KUKA KR120R2500 PRO industrial robot with a spindle end-effector. The KR120R2500 has a payload of 120 kg and reach of 2.5 m. It is controlled by a KUKA KR C4 controller.

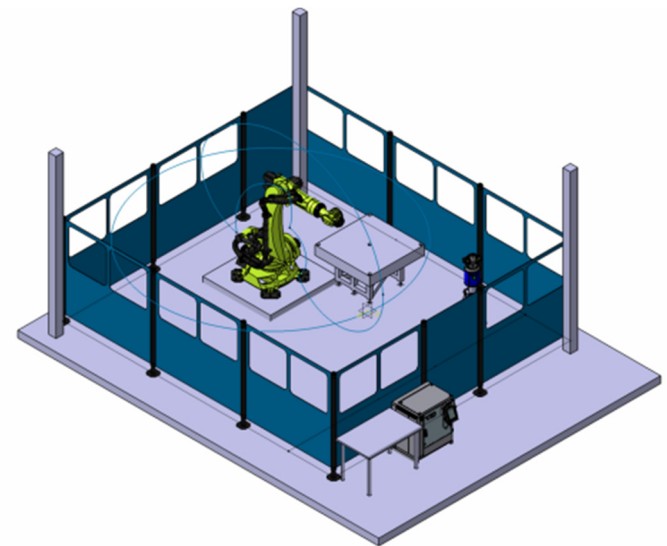

**Figure 1.** Laboratory for Integrated Metrology Applications (LIMA) robotic cell.

The spindle was a 400-V, 1-kW electric three-phase AC motor with ER20 tool holders (Figure 2), driven by a variable frequency drive. It was capable of 24,000 RPM, producing a maximum of 0.5–0.4 Nm of torque depending on RPM. The spindle had a run-out specification of 10 μm.

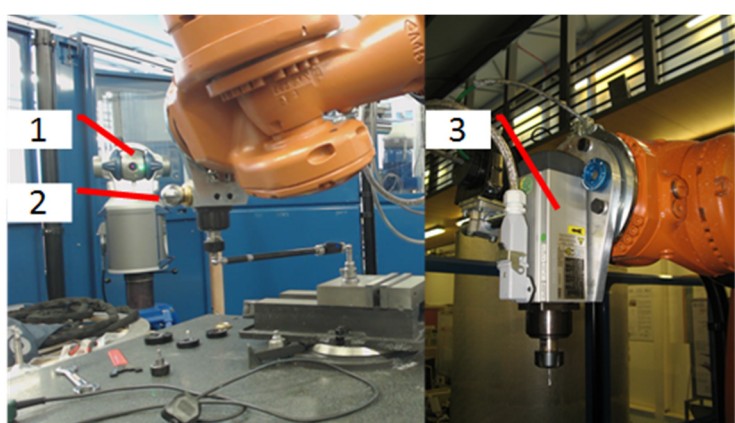

**Figure 2.** Laser tracker and spindle. (**1**) FARO laser tracker; (**2**) laser tracker spherically mounted reflector (SMR); (**3**) machining spindle.

The laser tracker used to compensate the 3-axis actuator was an absolute distance measurement (ADM) only FARO ION (Figure 2). It had a maximum permissible error (MPE) of 10 μm + 0.5 μm/m for distance [19]. It had a maximum update rate of approximately 860 Hz and was set to a more stable 512 Hz for the experiments. For a more detailed explanation of the working principles of the laser tracker, the reader could refer to the following references by Ester et al. [20], Peggs et al. [21], as well as the ASME performance verification standards [22].

## 3. Estimation of the SMR and TCP Offset

Spherically mounted reflector (SMR) and tool center point (TCP) offsets are crucial parameters required by the robot controller and the real-time feedback control process. Since the offsets affect the overall robot accuracy, they must be estimated accurately. There are many methods to determine these offsets. Typically, they are calculated by the robot controller while the operator moves the robot to visually align against a static reference object (usually a sharp pin). More accurate methods are used in

the offline calibration process, in which the robot is moved through a series of poses while measured by external instruments, and the offsets are solved automatically as unknowns along with the rest of the robot kinematic parameters.

In the case where the robot is compensated in real time by an external instrument, accurate knowledge about the robot kinematics is not required. A simple and repeatable process was developed to estimate the offsets from the SMR or TCP to the robot flange frame (Figure 3), using only the 5th and 6th (A5 and A6) axes of the robot. It was determined experimentally that the A5 and A6 are very repeatable, possibly better than laser tracker measurement uncertainty, thus they can be used as references themselves.

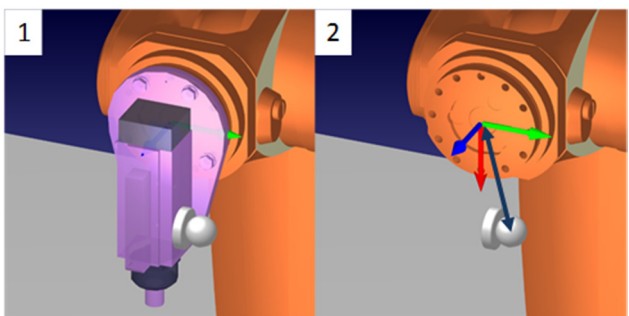

**Figure 3.** (**1**) SMR mounted on spindle; (**2**) SMR offset to robot flange frame.

The process starts with the robot at its home position, with a laser tracker SMR mounted to the side of the spindle end-effector. The axis A6 is then rotated through a full rotation while the SMR position is measured (Figure 4(1–4)). The initial position (A6 at 0°) is recorded as $P_0$ (Figure 5). Axis A5 is then rotated up and down as much as possible (at least ± 90°) while the SMR position is again measured (Figures 4–6).

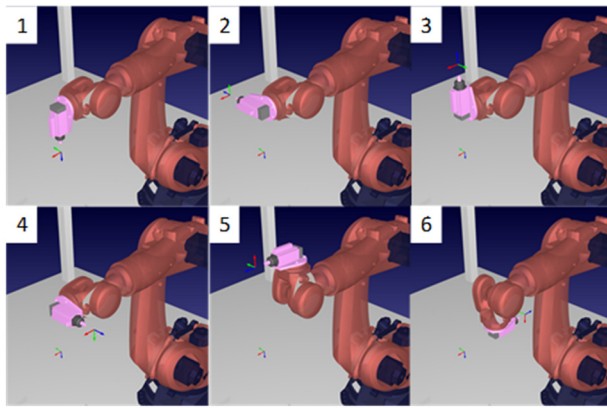

**Figure 4.** (**1**–**4**) Rotation about A6 axis to establish A6 rotation center; (**5**–**6**) rotation about A5 axis to establish A5 rotation center.

The reconstruction of the reference frame at the robot flange, has several steps. First circles can be best fit to the measurements from A5 and A6 rotations, creating positions $P_{A5}$ and $P_{A6}$ and unit normal vectors $\vec{n_{A5}}$ and $\vec{n_{A6}}$ (Figure 5). Note that we assume that $\vec{n_{A5}}$ and $\vec{n_{A6}}$ are approximately orthogonal, but not necessarily intersecting. We note that the unit basis vectors of the reference frame can be created from the circle normal vectors:

$$\hat{n}_Z = \vec{n_{A6}} \tag{1}$$

$$\hat{n}_X = \vec{n_{A5}} \times \vec{n_{A6}} \tag{2}$$

$$\hat{n}_Y = \hat{n}_Z \times \hat{n}_X \tag{3}$$

Therefore, the rotation matrix $R$ from the laser tracker instrument frame to the robot flange frame is:

$$R = \begin{bmatrix} \hat{n}_{X'} & \hat{n}_{Y'} & \hat{n}_{Z'} \end{bmatrix} \tag{4}$$

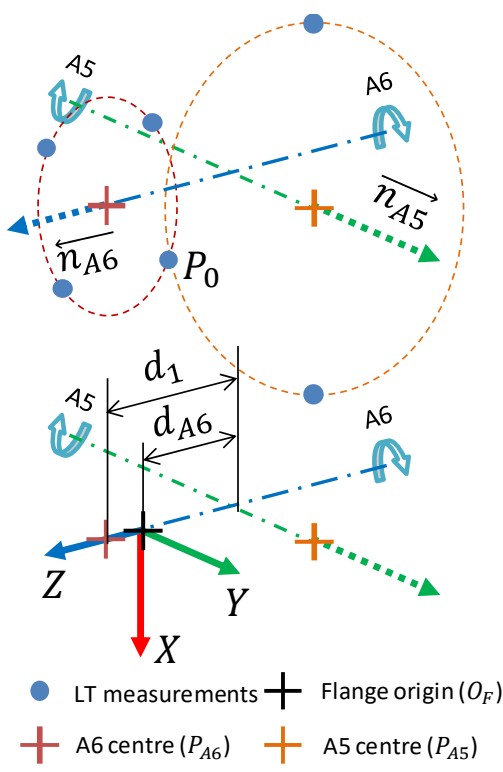

**Figure 5.** Reconstruction of the robot flange reference frame.

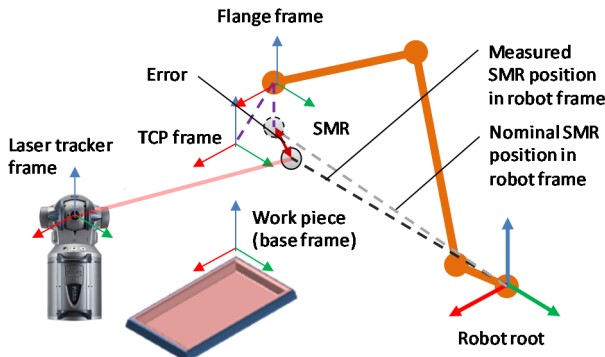

**Figure 6.** Coordinate frames involved in real-time compensation.

To calculate the position of the flange origin $O_F$, the distances $d_1$ and $d_{A6}$ must be known. It is possible to simply use the nominal distance $d_{A6}$ found the robot manual, since any small errors in its length will result in a position error that will be easily compensated by the laser tracker feedback. The distance $d_1$ on the other hand can be calculated using scalar projection of the $P_{A5}$ and $P_{A6}$ vector to the Z basis vector:

$$d_1 = \overrightarrow{P_{A6}P_{A5}} \cdot \hat{n}_Z \tag{5}$$

The flange origin $O_F$ in laser tracker reference frame is therefore:

$$O_F = P_{A6} - (d_1 - d_{A6}) \cdot \hat{n}_Z \tag{6}$$

With origin $O_F$ and rotation matrix $R$, it is now possible to calculate the position of $P_0$ in the robot flange frame, i.e., the SMR offset $P_{SMR}$:

$$P_{SMR} = R^{-1} \times (P_0 - O_F) \tag{7}$$

The TCP offset can be calculated in a similar way to the SMR offset, by using a tool such as a vector-bar holding two SMRs concentrically in the spindle chuck. In this way, the spindle orientation vector can also be calculated.

## 4. Coordinate Transformations

A common coordinate frame of reference is an essential requirement for real-time position compensation. While any reference frame can be used for this purpose, there are some complications and considerations in its choice.

Typically, offline programming software for industrial robots generates robot paths using the workpiece frame and the TCP frame, both of which are referenced to the robot root (Figure 6). The programs are generated with the assumption that the nominal kinematic model of the robot is perfect.

It was determined that if the compensation can be applied to in the robot root frame, it is possible to maintain the same familiar offline programming workflow, so that that process of the creating robot path with compensation is no different from the normal uncompensated process. This therefore requires the measurements by the laser tracker to be transformed into the robot root frame. While the transformation can simply be found by moving the robot to a few positions and best fit the tracker measurements to the nominal robot coordinates, there are a few technical complications.

First, in a robot program, once a tool is selected, the position of the TCP frame of the tool is reported by the robot controller. The TCP position, however, cannot be directly measured by the laser tracker. The laser tracker can only measure the position of the SMR, but the nominal position of the SMR (i.e., the robot controller's estimation of the SMR position) is not reported by the robot controller (Figure 6).

Second, although the SMR offset $P_{SMR}$ can be calculated using the method described previously, it is relative to the robot flange. The KUKA KRC4 controller unfortunately does not report the flange frame position once a tool is selected. The solution is to take the SMR offset $P_{SMR}$ through the inverse of the tool reference frame, which is derived using PSMR_TCP = INV_POS($TOOL):PSMR in KUKA robot language, producing the SMR position in the TCP frame $P_{SMR\_TCP}$. Then, in the real-time loop, transform $P_{SMR\_TCP}$ into the robot root frame through inverse Robot root → TCP frame transform. This transformation effectively cancels out the effect of the TCP → flange transform.

Apart from the aforementioned work flow advantages, another benefit of this method is that it allows the compensation to happen independently of the selected tool or work frames, such that real-time corrections can be easily and quickly applied to any robot program, including ones made without compensation in mind, without major changes.

## 5. Real-time Closed Loop Compensation

Closed loop compensation of the robot is achieved through the laser tracker measurement of a SMR attached to the robot end-effector. The hardware and data flow are shown in Figure 7. Similar methodology was tested successfully at a smaller scale on 3-axis machines in our previous papers [23,24].

This control method requires the robot TCP and the laser tracker to be in the same coordinate frame. In order to bring the two systems into the same frame, first, the offset of the laser tracker target reflector (SMR) and the spindle tool center point (TCP) relative to the robot flange must be known. The offsets are carefully calibrated using the laser tracker using the method described in Section 3.

Tests found mounting the SMR closer to the TCP increases accuracy, however, mounting the SMR too close to the cutting tool can result in flying swarf damaging the reflector mirrors.

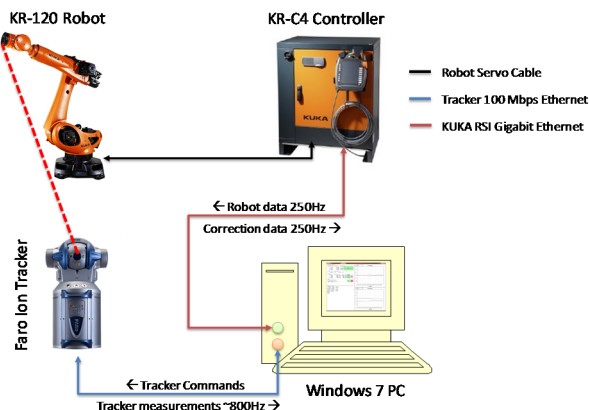

**Figure 7.** Hardware and data flow.

Once the TCP and SMR offsets are calibrated, it is possible for the robot controller to send its current estimated SMR position in the robot base coordinate frame to a PC through the kinematic chain, over the KUKA RSI (robot sensor interface).

The robot can be moved to different positions and both the laser tracker measurement of the SMR in the tracker frame and the robot estimated SMR position in the robot frame can be recorded. A coordinate transformation between the two frames can be calculated with a least-squares fitting of the two sets of measurements, as described in Section 4. This transformation can then be used to generate correction vectors in the robot frame from tracker measurements.

*5.1. Control Loop*

Control software was developed at Bath (Figure 8). The software runs on a Windows PC, handling communications with the laser tracker over ethernet through the FARO laser tracker software development kit (SDK). The software also communicates with the KUKA controller through KUKA robot sensor interface (RSI), sending and receiving data every robot controller cycle. A proportional-derivative (PD) controller is implemented in the control software (Figure 9).

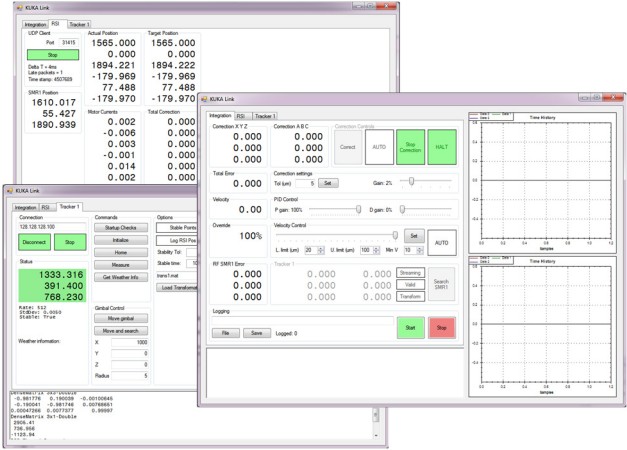

**Figure 8.** PC control software developed for real-time compensation.

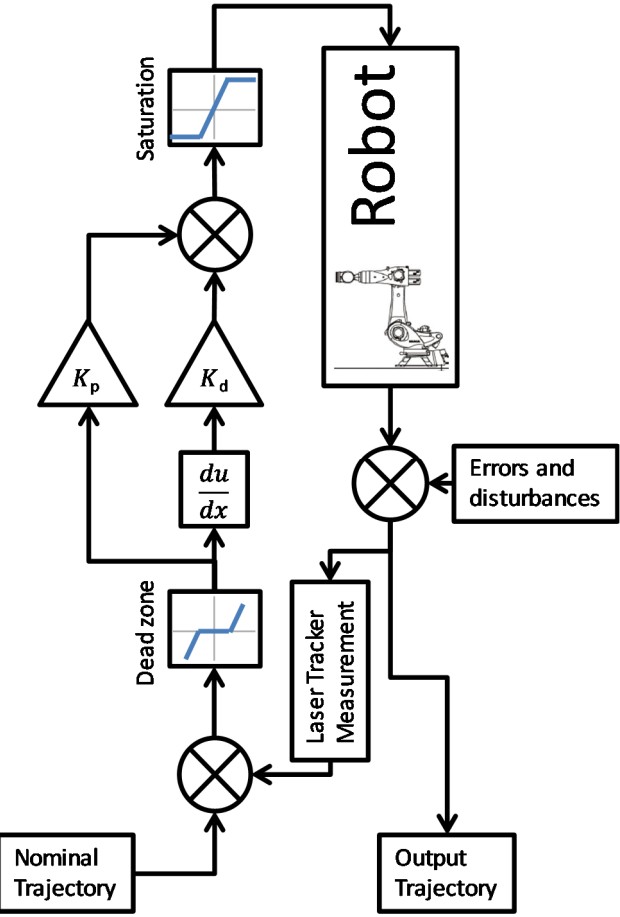

**Figure 9.** Main controller loop.

During the real-time process, every 4 milliseconds the robot sends its estimated position of the SMR in the robot frame to the controller software. This position is compared to the latest laser tracker measurement transformed into the robot frame. If the error between the two positions is greater than a preset threshold, a correction vector is generated and sent to the robot controller. The threshold is typically set at 20 μm, to stop the effect of laser tracker noise on the control loop. PD gains as well as saturation limits are applied to the correction vector to further tune the control loop. Note that the correction vectors sent to the robot controller adjusts the set point of the controller, rather than adjusting lower level states such as motor torque and the effect of the corrections are cumulative, therefore the overall effect of the controller is similar to that of a PI control.

*5.2. Position Correction vs. Path Correction*

Since the robot controller and the laser tracker have independent clocks and update frequencies, it is not possible to perfectly synchronize the laser tracker measurement with the robot control loop. This results in jitter in the measured error vector in the direction of the robot movement. The error caused by synchronization jitter is worse at higher movement speeds and can lead to unstable oscillations in the control loop. Additionally, since the error created by jitter does not affect the path accuracy, attempts to compensate for it wastes control effort.

A solution to this problem was found by defining the error vector as the error between the LT measured position and the robot motion trajectory rather than the current robot position, such that synchronization errors do not affect the error vector (Figure 10).

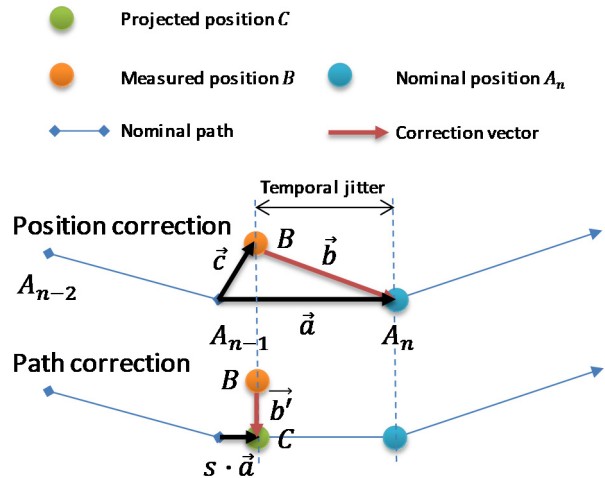

**Figure 10.** Position error vector ($\vec{b}$) vs. path error vector ($\vec{b'}$).

Consider the SMR position in the robot coordinate frame reported by the robot controller moving through a nominal trajectory $A$ consisting of a series of positions:

$$A = \{A_n, A_{n-1}, A_{n-2}, \cdots\} \tag{8}$$

where $A_n$ is the position of the most recent time step and $A_{n-1}$ is the position at the previous time step. Therefore, the motion vector $\vec{a}$ is simply:

$$\vec{a} = A_n - A_{n-1} \tag{9}$$

If position $B$ is the laser tracker measured position of the SMR in the robot frame, then the vector projection of $B$ in vector $\vec{a}$ creates the point $C$:

$$C = A_{n-1} + s \cdot \vec{a} \tag{10}$$

where:

$$s = \vec{c} \cdot \frac{\vec{a}}{\|\vec{a}\|} \tag{11}$$

$$\vec{c} = B - A_{n-1} \tag{12}$$

The path error vector $\vec{b'}$ is then

$$\vec{b'} = C - B \tag{13}$$

Note that when the velocity of the robot approaches zero, Equation (11) approaches infinity. It is therefore necessary to switch back to position correction when robot speed is low. In the robot control PC software this cut-in speed is set at 1 mm/s. At 1 mm/s, if the laser tracker data are delayed by a full robot controller cycle of 4 ms, the maximum jitter error is only 4 μm, small enough to be negligible for control stability. The value of $s$ in Equation (11) is also clamped to $0 < s < 1$ in the control software for additional robustness.

To demonstrate the effect of the using path correction vs. position correction, the robot is programmed to move in a square trajectory at 20 mm/s. The error vectors for the trajectory are recorded for four test cases: no feedback, P feedback, PD feedback and finally, PD feedback using path correction.

Figure 11 shows that using position correction with proportional control, there is significant jitter noise. The addition of derivative control allows the robot to react faster to sudden changes in error,

reducing eliminating some of the larger spikes, but jitter noise remains. PD control using path error vector instead of position error mostly eliminated jitter noise, allowing much higher control gains to be used, with no limitation on robot motion speed.

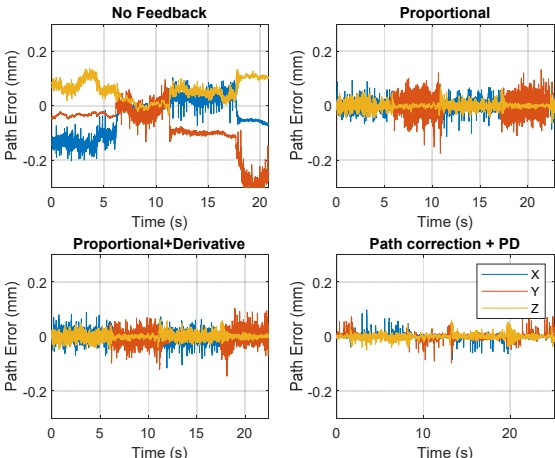

**Figure 11.** Error vector vs. time for a square trajectory demonstrating the effect of using path correction vs. position correction in jitter noise reduction.

While ideally the measurement instrument should be perfectly synchronized with the robot controller, this may not always be possible, as many metrology systems are not designed for the purpose of real-time robot control. Path correction makes it possible for instruments with much wider ranges of update frequencies and measurement latencies to be used for robot path compensation.

## 6. Ballbar Tests

The Renishaw ballbar (Figure 12) is a linear measurement device that is typically used for quick machine tool performance checks and the diagnosis of potential problems, such as axis scale, squareness and backlash.

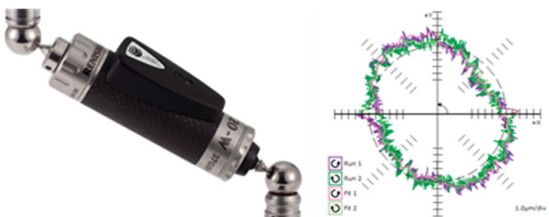

**Figure 12.** Renishaw QC20-W ballbar and typical test results [25].

It consists of two precision tooling balls attached to a telescoping linear transducer. Two kinematic mounts, one placed on the machine bed and one attached to the spindle, holds the ballbar in place as the machine performances a series of programmed circular paths about the center of one of the spheres. The distance between the two balls (deviation from the nominal circle) is recorded during the operation and can then be used to analyze the accuracy of the machine tool according to ISO 230-4, ASME B5.54 standards or Renishaw's proprietary diagnostic tools.

The QC20-W ballbar used in the study has a resolution of 0.1 μm and measurement uncertainty of 0.7 μm + 0.3% L (i.e., for a 100 μm measured circular deviation the uncertainty is 1 μm) [25]. The ballbar can make dynamic measurements and is at least an order of magnitude better precision than a laser tracker. Hence, for the purposes of the experiments described in this study, it is a very useful tool to independently validate the dynamic performance of the system. The ballbar also measures from the

end of the tool, which is offset from the SMR, such that it will be able to record any orientation errors, which cannot be compensated by the laser tracker.

## 6.1. Ballbar Length Calibration

In order to demonstrate the full capabilities of the system over a larger volume and the improvement in the software and control algorithms since the initial tests, a series of 600-mm radius ballbar tests were conducted at various speeds.

The ballbar is normally used with a pre-calibrated scale artifact, which has set lengths of 50, 100, 150 and 300 mm. In order to increase the measurement radius to 600 mm, a custom length scale standard was constructed on a low thermal expansion National Physical Laboratory (NPL) length artifact using 0.5-inch SMR nests. A Leica 401 laser tracker was used to measure the distance between the nests, using mostly the distance component of the tracker (Figure 13).

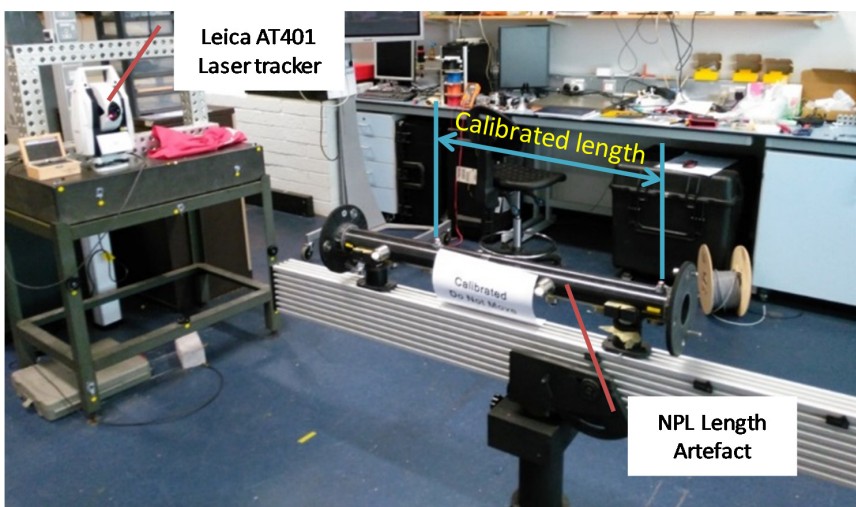

**Figure 13.** 600-mm-length reference measured with laser tracker.

Over 200 measurements were made over a 2 h period. The mean distance is 600.0685 mm with a standard deviation of 3.5 μm.

Using the National Physical Laboratory laser tracker simulation code [26,27] and the published specifications for the Leica 401 tracker [28], it is possible to simulate the measurement uncertainty of the distance between the 2 measured nests (Figure 14) used for the calibration of the ballbar. The measured length uncertainty is ±7.3 μm $k = 1$.

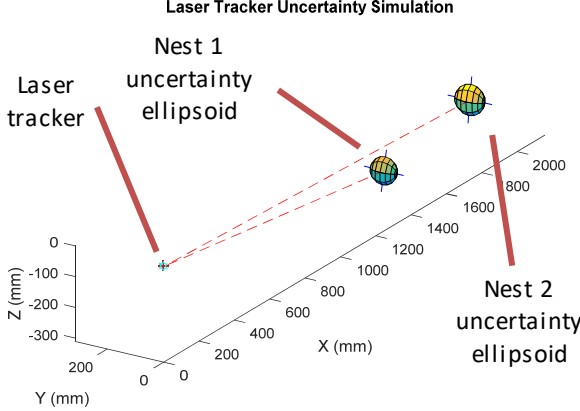

**Figure 14.** Laser tracker uncertainty simulation of ballbar calibration length.

## 6.2. Experiment Setup

Each experiment starts with the calibration of the ballbar on the NPL artifact (Figure 13). Each run consists of two counterclockwise (CCW) and two clockwise (CW) circles with a 4 s pause in between (Figure 15). The center of the circle is approximately 1.1 m above the floor mounted on a tripod (Figure 16), and the circle is tilted 45° away from the robot.

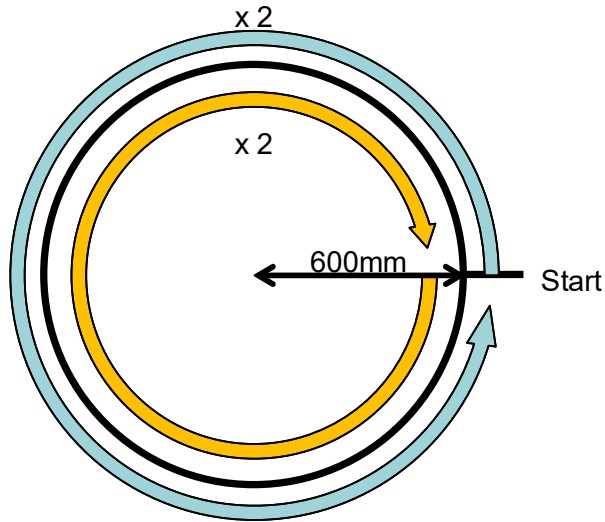

**Figure 15.** Programmed path for ballbar test.

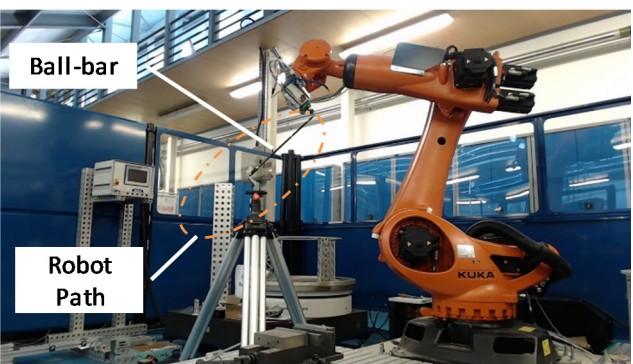

**Figure 16.** Ballbar test in progress.

The complete experiment consists of three no-feedback runs and three with feedback runs at three different feed rates, selected from typical machining speeds. Table 1 lists all runs completed.

**Table 1.** List of all ballbar test runs.

| Feed Rate (mm/min) | With Feedback | Number of Runs |
|---|---|---|
| 1000 | NO | 3 |
| 1000 | YES | 3 |
| 500 | NO | 3 |
| 500 | YES | 3 |
| 250 | NO | 3 |
| 250 | YES | 3 |

### 6.3. Ballbar Test Results

The results from one of the three runs for each feed rate is shown in Figure 17. The differences between the three runs are very minor. Without compensation, the robot trajectory tends to be at a larger radius, with significant backlash errors approximately 65°, 175°, 280° and 350°. With compensation, the errors are mostly eliminated.

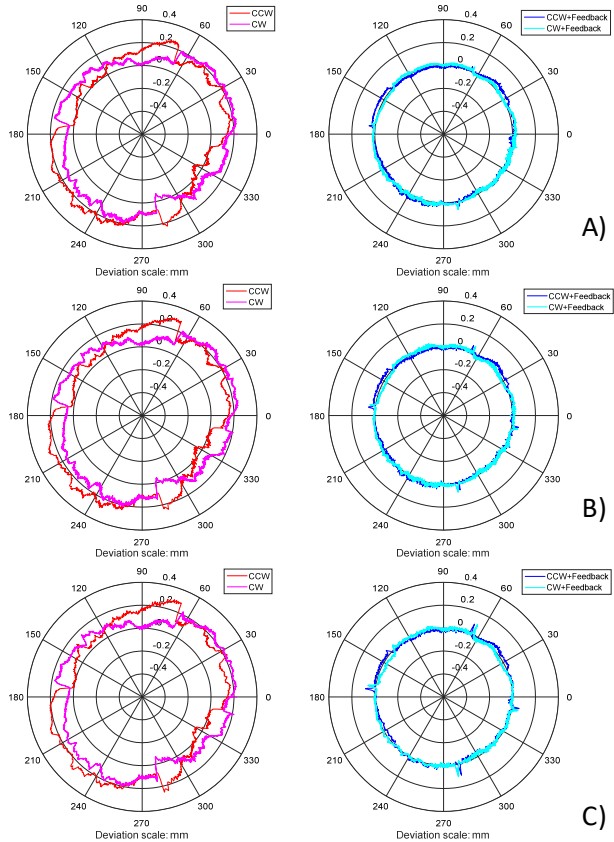

**Figure 17.** Ballbar results for feed rates for one run. (**A**) 250 mm/min; (**B**) 500 mm/min; (**C**) 1000 mm/min.

Best-fit circle radius error are summarized in Table 2, and the RMS errors of all the test runs are plotted in Figure 18. RMS errors reduced from around 140 μm to mostly less than 20 μm with compensation enabled. The 95th percentile error is reduced from 240 μm to 35 μm. There were no clear effects feed rate on the overall trajectory accuracy, apart from increases in reversal spikes. Although the highest robot speed tested is 1000 mm/min, representative of typical machining speeds, the robot capable of moving much faster. It is likely that at even higher speeds the RMS errors will increase.

**Table 2.** Best-fit circle radius error comparison (mm), ballbar calibration uncertainty: ±7.3 μm $k = 1$.

| Run Type | 1000 mm/min | 500 mm/min | 250 mm/min |
|----------|-------------|------------|------------|
| No FB    | 0.1245      | 0.1366     | 0.1202     |
| No FB    | 0.1319      | 0.1155     | 0.1382     |
| No FB    | 0.1347      | 0.1219     | 0.1160     |
| With FB  | −0.0032     | 0.0045     | 0.0093     |
| With FB  | −0.0042     | 0.0047     | 0.0081     |
| With FB  | 0.0013      | 0.0043     | 0.0026     |

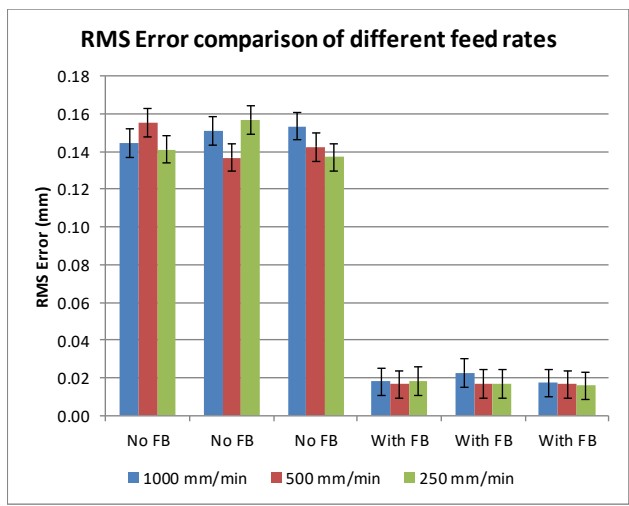

**Figure 18.** RMS error of ballbar test for different feed rates for all three runs. Error bars for ballbar calibration: ±7.3 μm *k* = 1.

RMS errors represent the overall position accuracy of the robot relative to the center of the ballbar. It is a statement of the improvement of absolute accuracy achieved, including any errors from the calibration of the TCP, setting up of the ballbar center point and coordinate transformations.

By best-fitting a circle to the ballbar results and comparing the radius of the best-fit circle to the nominal circle, it is also possible to highlight the form error of the circles. Table 2 shows that with feedback-fit circle radius errors are reduced to well below 10 μm, while without feedback the circle radii tend to be more than 110 μm larger than nominal.

It is important to note that although the ballbar is a useful instrument to analyze dynamic trajectory accuracy of robots, it is only a 1D measurement system. It cannot by itself characterize a robot's 3D accuracy since any errors normal to the ballbar test plane are not recorded.

## 7. Block Drilling Tests

A series of drilling trials were conducted to test the performance of the robot with and without real-time compensation (Figure 19). The trials where designed to replicate drilling of aerospace components on behalf of an aerospace company.

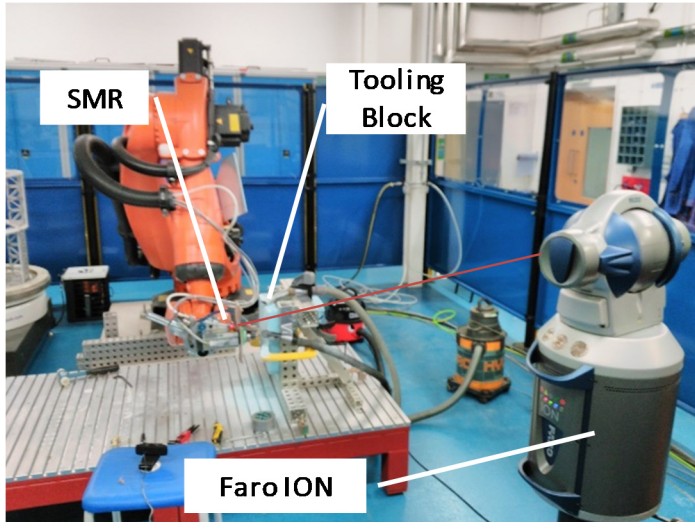

**Figure 19.** Picture of robot drilling test block under real-time laser tracker compensation.

*7.1. Experiment Setup*

The trials involved the drilling of a set of 25 holes in a 5 × 5 pattern with 10 mm spacing between the holes. The trials were repeated at 4 different robot positions in order to test the robot performance at various joint configurations as shown in Figure 20.

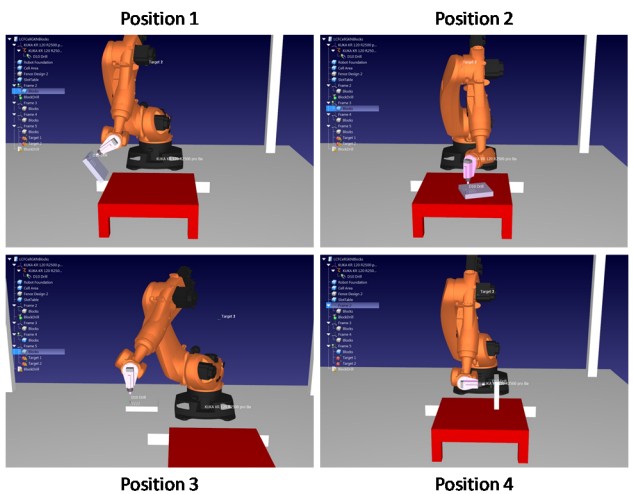

**Figure 20.** Robot configurations tested.

At each position, the one set of holes are drilled while under real-time compensation, while the other set is the control set with no compensation. Before each trial, the 6-DOF positions of the blocks were measured using the laser tracker, by probing the block faces directly. The block positions were then converted into the robot base coordinate system using the transform described in Section 3. Robot programs for the drilling tool paths are then generated using these block positions. The drilling parameters for the trials are:

- Material: Epoxy tooling blocks;
- Drill: 10 mm 2 flute carbide twist drill;
- Spindle RPM: 6500;
- Coolant: none;
- Feed rate: 240 mm/min;
- Hole depth: 35 mm.

*7.2. Block Drilling Results*

The drilled blocks are measured on a coordinate measurement machine (CMM) with a point measurement uncertainty of 5 μm ($k = 2$). Each hole was probed with 7 circles with depth intervals of 5 mm, 16 points per circle. Figure 21 shows an example of the results, taken from position 2.

Under real-time laser tracker feedback, the holes in general have lower median radial errors, improving 30%–50% except for position 3, where the radial errors without feedback being slightly better. This is likely caused by the position being higher up, with more compliance within the holding fixture, allowing the block to move to self-center to during drilling. Some anecdotal evidence from another robotics research group at the University of Sheffield AMRC also suggested better hole quality as a result of more compliant part holding. There is, however, no detailed study to confirm this effect.

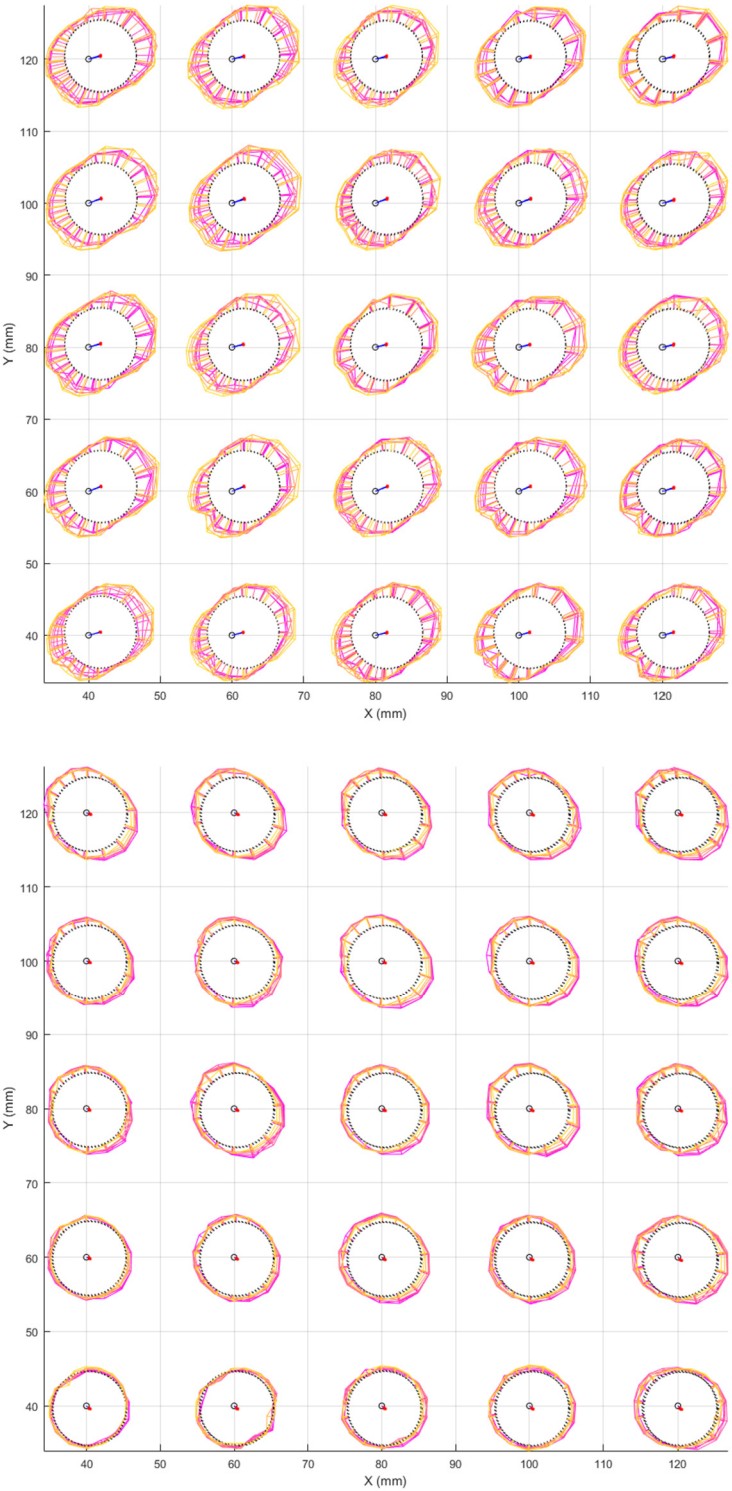

**Figure 21.** Results from position 2:20× radial error exaggeration, 2× hole position exaggeration. (**top**) without compensation; (**bottom**) with compensation.

The positions of the best-fit hole centers are shown in Figure 22 and the mean hole center positions errors are summarized in Figure 23. Under feedback, the absolute positions of the drilled holes are significantly improved. Without feedback, the hole positions errors can be as much as 0.83–0.45 mm, with feedback the errors are reduced to between 0.17–0.10 mm. Under feedback, the errors are all in approximately the same direction, likely caused by a datum shift from the calibration process

or differences between the how the block is probed on the CMM and in robot cell. These kinds of consistent errors are easy to calibrate out, further improving performance.

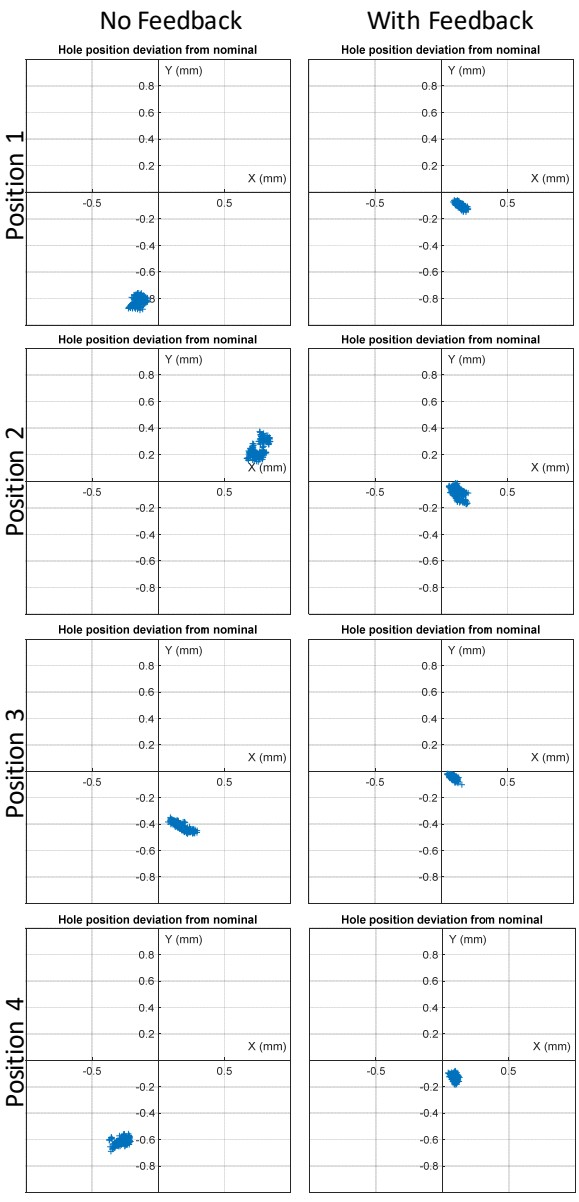

**Figure 22.** Best-fit hole center absolute position errors for all positions.

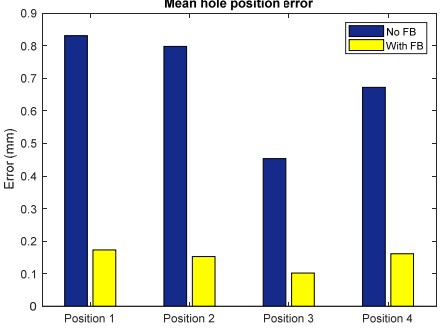

**Figure 23.** Summary of hole position error of all 4 positions.

If the datum shift is removed by subtracting the mean position error of all 4 positions (Figure 24), the remaining position errors range from 0.036 to 0.051 mm, with 95% of all hole centers within 0.081 mm. Additionally, the spreads of the hole centers are also smaller when feedback is enabled, 15%–50% reduction of hole center position standard deviation can be observed.

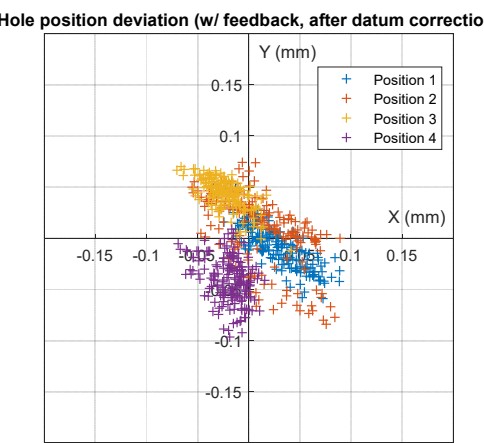

**Figure 24.** Best-fit hole center absolute position errors for after datum shift is removed.

While the hole positions improved significantly, due to the lack of any feedback on the rotation of the end-effector, the normalities of the holes relative to the block surface are not improved (Figure 25) and can be worse than without feedback.

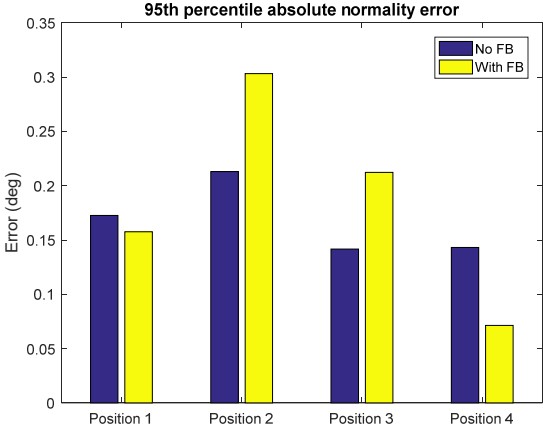

**Figure 25.** Summary of 95th percentile hole normality for all 4 positions.

## 8. Aluminum Machining Experiment

Several aluminum parts were machine by the robot. These involved interoperability of the robot and laser tracker systems with additional metrology instruments provided by our collaborators at Loughborough University and University College London.

This section presents only the results from the real-time laser tracker compensation. For more details on other technologies tested, including the fringe projection scanning system and object recognition algorithms developed by Loughborough, the low-cost photogrammetry system developed by UCL and the active robot vibration control system developed at Bath please refer to the references [29–31].

An aluminum component representative of a typical aerospace component used in wing spar assemblies was designed. The component is 230 × 110 mm, machined out of a blank 300 × 200 mm EN AW-2017-T351 aluminum plate using a 10 mm two-flute carbide endmill cutter. The machining time

was 1 h 20 min. Two of the components were made using the robot, one with real-time compensation, one without. The components were measured on a CMM and compared against nominal CAD geometry (Figure 26).

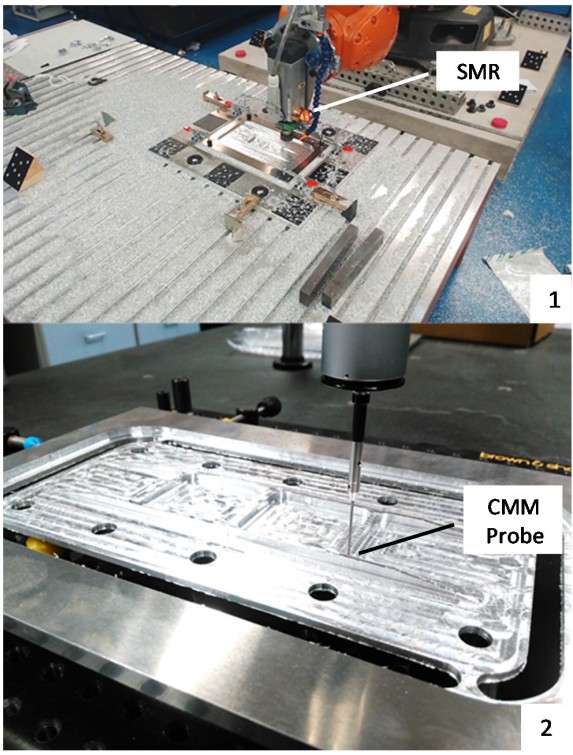

**Figure 26.** (**1**) Machining of the part and (**2**) CMM measurement of the finished part.

Note that in this experiment, only the robot's accuracy in a small volume is tested. The results do not include the absolute accuracy of the robot, which can be much worse without compensation as shown in the previous section on drilling.

Figures 27–29 present graphically the CMM results of different features of the part, with colored scales from −240 μm to +240 μm.

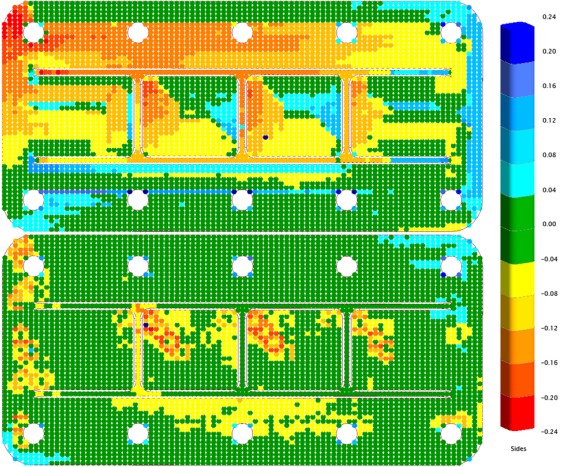

**Figure 27.** Comparison of measured main plane vs. nominal CAD. (**top**) without feedback; (**bottom**) with feedback.

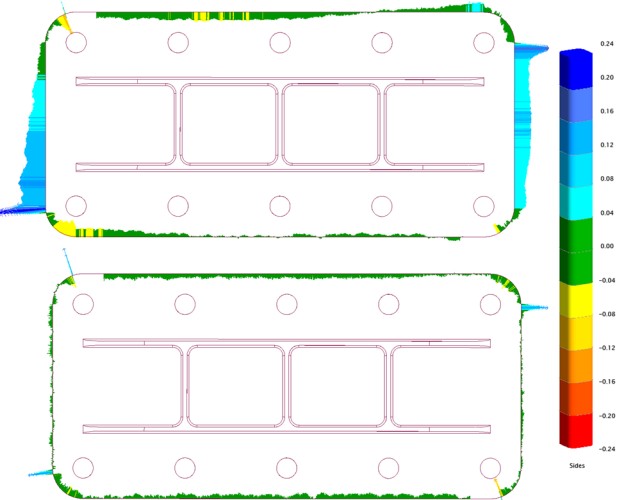

**Figure 28.** Comparison of measured periphery of the part vs. nominal CAD. (**top**) without feedback; (**bottom**) with feedback. Errors are indicated in mm.

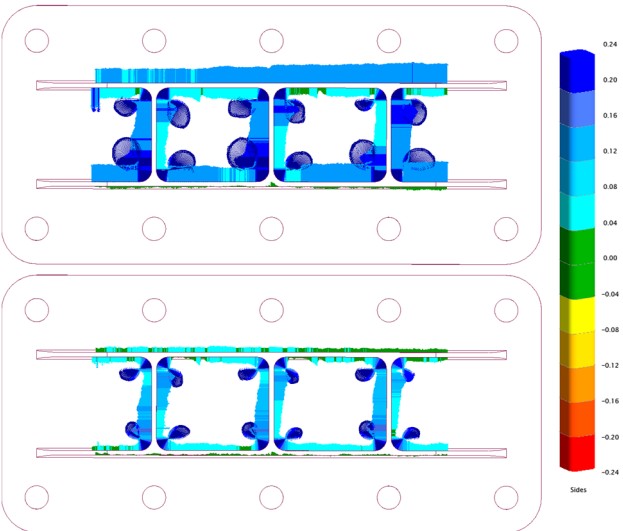

**Figure 29.** Comparison of measured stiffener profile vs. nominal CAD. (**top**) without feedback; (**bottom**) with feedback. Errors are indicated in mm.

Measurement results of the main plane features of the part and the tops of the stiffeners are shown in Figure 27. Without compensation, large portions of the plane and the entire stiffener tops are above ±40 µm from nominal. Only small parts of the planes have errors above ±40 µm when compensation is enabled.

Figure 28 shows the profiles of the periphery of the part compared to nominal CAD. Again, when compensation is enabled, almost the entire profile is within ±40 µm apart from some spikes near the corners, likely caused by robot joint backlash.

The stiffener profiles are shown in Figure 29. While real-time compensation significantly improved the accuracy of the feature, the inside corners of the pockets are not very well machined. The conventional pocketing toolpath used likely led to sudden increase in cutter engagement at the corners. The robot may not be able to compensate for the sudden increase in force required in time, even with laser tracker compensation. Increased vibration was also observed during machining of the stiffeners, likely a result of the thin features and the part having poor mechanical support towards the middle.

The CMM results comparisons of the features are also summarized in histogram and table form in Figure 30 and Table 3.

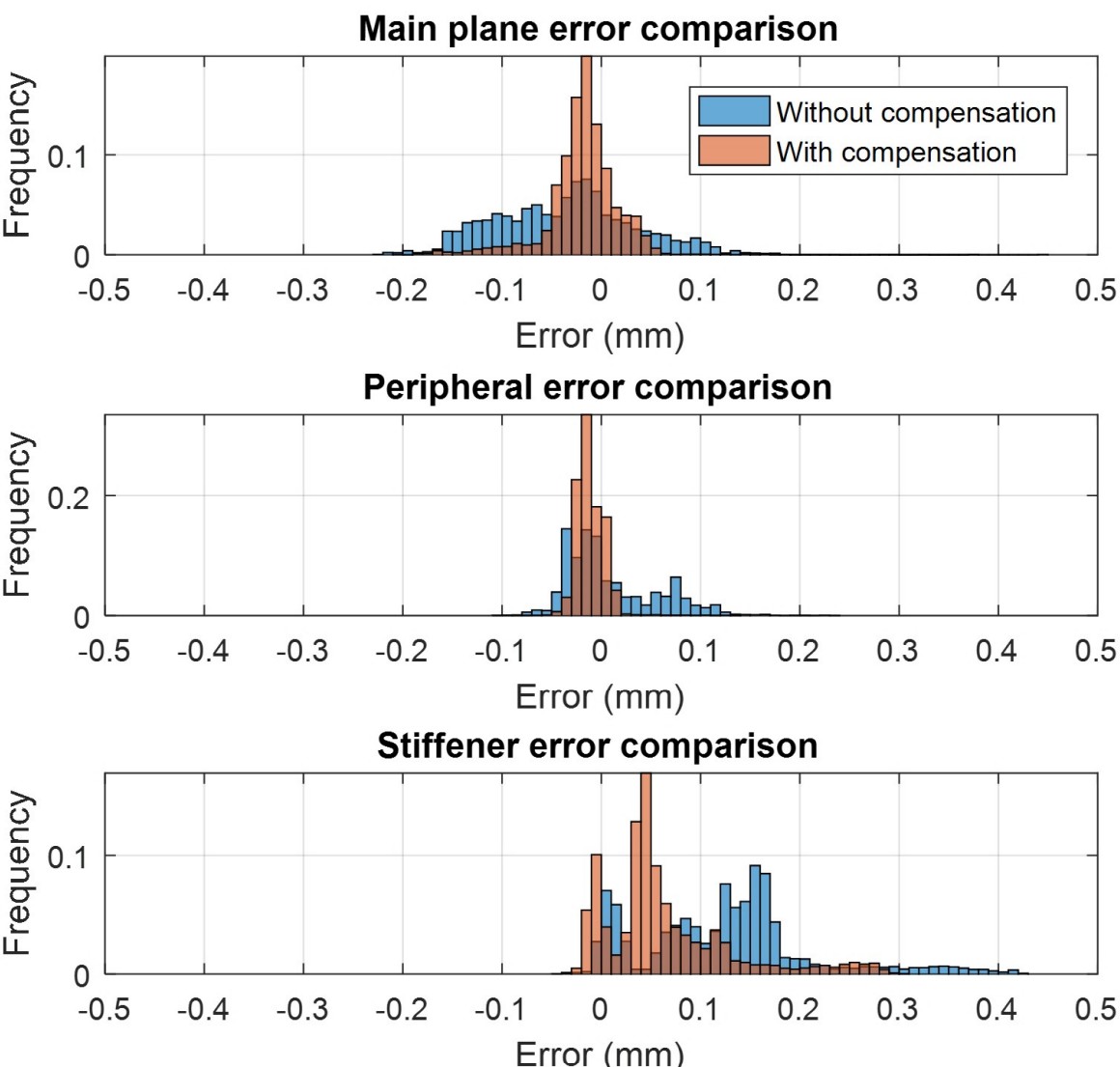

**Figure 30.** Histogram comparison of feature accuracy.

**Table 3.** Summary table of percentage of feature within ±50 μm.

| Feature | With Feedback | Without Feedback |
| --- | --- | --- |
| Main Plane | 88.6% | 46.3% |
| Stiffener | 99.1% | 74.8% |
| Periphery | 54.9% | 19.7% |

Figure 30 shows that with compensation, both the median and the variance of the features are significantly improved. If a ±50 μm form tolerance is applied to all the features of the part, Table 3 shows the percentages of the features that would pass this tolerance requirement. With feedback, 88.6% of the main plane, 99.1% of the periphery and 54.9% of stiffener features of the part are within ±50 μm of nominal CAD, an improvement of 1.9×, 1.3× and 2.8× compared to without feedback.

## 9. Robot Backlash Investigation and Automatic Speed Control

Results from the ballbar tests and machining experiment show that while the accuracy of the robot under feedback is improved, there are some areas where significant errors can occur. These sudden error spikes are on the order of several hundred microns and are large enough to ruin a machined part even though the rest of the part is within specifications. These spikes are theorized to be caused by joint backlash. They typically occur during reversals of one or more robot joints.

Although the feed rates tested had no major effect on the overall RMS errors during ballbar tests described in Section 6, Figure 31 highlights the effect of feed rate on the reversal spike occurring around 295° in the clockwise direction (65° CCW in Figure 17). As speed increased, both the amplitude and the duration of the spike became larger.

Figure 32 shows the CMM measurement result from with the machining tool path from Section 8 overlaid on top, illustrating graphically that the areas of largest errors coincide with sudden changes in toolpath direction.

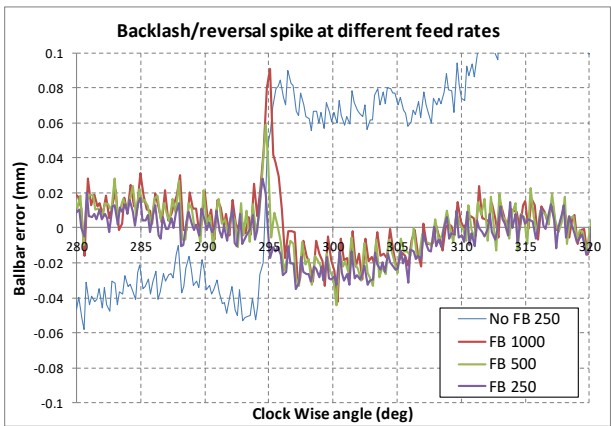

**Figure 31.** Comparison of ballbar test backlash/reversal spike occurring at around 295° clockwise (65° CCW in Figure 17) for different robot speeds.

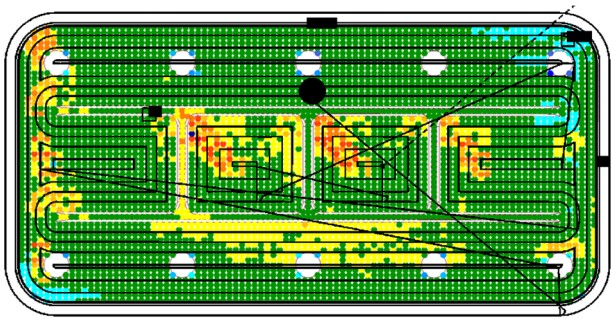

**Figure 32.** Tool path overlaid on CMM result, showing correlation between areas of large error with sudden change in toolpath direction.

Joint backlash causes a sudden spike in position error, which is not completely removed through feedback control, because the control loop can only react to a measured error. By the time a spike is measured, it is already too late to compensate for it.

While ideally backlash should be compensated using a model-based controller, possibly with feedforward control, an attempt was made to reduce the effect of backlash by overriding the robot motion speed while under laser tracker closed loop feedback.

A control logic was added to automatically reduce robot speed when a large error is measured, therefore giving the control loop enough time to correct the error, before ramping back to the original

speed. The speed reduction is applied to the robot velocity override variable and is proportional to the total path error as measured by the laser tracker, with a lower limit of 20%. A gain of 1000 was used, such that a 100 μm error leads to 100% reduction in speed (clamped to 20%). A slow ramp up of 0.25% per time step (4 ms) is applied to minimize sudden acceleration and stuttering when the error is corrected, and the robot immediately resumes full speed.

To test the effect of automatic speed control (ASC), three aluminum test tokens were machined with the robot and areas of backlash error were measured on a Taylor Hobson surface profiler (Figure 33) using an inward helical tool path. The first test token is machined without any feedback, the second token is machined with feedback and the last one is machined with feedback and automatic speed control.

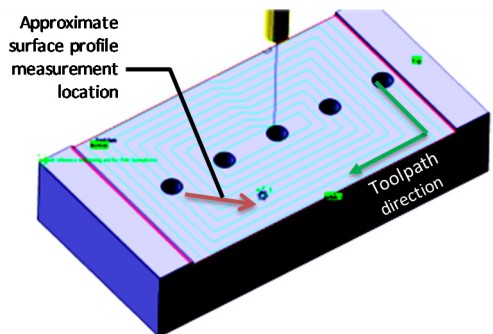

**Figure 33.** Backlash test token showing toolpath and position of surface profile measurement.

Figure 34 shows the result of the surface profile measurements, note while the profile is measured left to right, the cutting tool is moving from right to left.

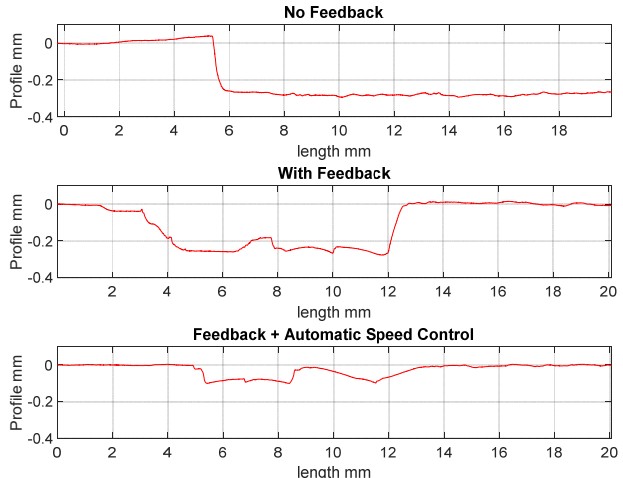

**Figure 34.** Surface profile measurement comparison.

Without feedback, the large spike of error essentially becomes a step, leading to geometric errors in a large portion of the part. With feedback, the error is corrected after 11 mm, but a gash is cut into the part at a depth exceeding 250 μm. With automatic speed control, the max depth deviation is reduced to 100 μm, and recovery occurred after approximately 8 mm.

Although automatic speed control reduces the effect of backlash significantly, there are, however, several important disadvantages to ASC. First, it increases considerably cycle time for toolpath that involve many direction reversals. Second, it leads to uneven cutter chip load and variation in surface finish as the feed rate varies.

Backlash remains as the largest source of residual error for a robot under real-time metrology compensation. To achieve results similar to a normal machine tool, future work on the analysis and modeling of robot joint backlash is required.

## 10. Summary and Conclusions

A detailed description of the hardware and software real-time robot compensation setup is described in this study. The TCP calibration process, coordinate frames, and control loops and the control algorithms are also explained.

To test the performance of the robot, three sets of experiments were conducted, include ballbar dynamic path accuracy test, a series of drilling case studies and a machining test.

Ballbar tests of the robot's dynamic accuracy showed that the RMS errors reduced from around 140 μm to mostly less than 20 μm with compensation enabled. The 95th percentile error is reduced from 240 μm to 35 μm. There were no clear effects feed rate on the overall trajectory accuracy, apart from increases in reversal spikes. It also demonstrated a 10× improvement in the best-fit circle radius.

Results of robotic drilling under real-time laser tracker compensation are presented in this study. Reductions of hole position errors from 0.83–0.45 mm to 0.17–0.10 mm can be observed, even though the feedback only contained 3-DOF position information. This result can be further improved to below 0.051 mm after removing datum shift because of the consistency of the errors under feedback. The standard deviations of the hole positions are also improved. As expected, the hole normality is not improved, due to the lack of rotation information.

In the machining test, two aluminum component representative of a typical aerospace component used in wing spar assemblies were machined by the robot. Under real-time laser tracker feedback, the proportion of the main plane, the periphery and the stiffener features of the part within ±50 μm tolerance improved 1.9×, 1.3× and 2.8× compared to without feedback.

Finally, the effect of robot backlash observed during the ballbar tests and machining experiments were discussed. It was shown that real-time metrology feedback cannot fully compensate for the sudden error spikes caused by backlash. The mitigation strategy of reducing feed rate automatically (ASC) was demonstrated to significantly reduce backlash error. However, ASC considerably increases the cycle time for a toolpath that involve many direction reversals and leads to uneven cutter chip load and variation in surface finish. Backlash therefore remains as the largest source of residual error for a robot under real-time metrology compensation, requiring detailed analysis and modeling in the future.

**Author Contributions:** Conceptualization, Z.W.; Data curation, Z.W and R.Z.; Funding acquisition, P.K.; Investigation, Z.W.; Methodology, P.K.; Project administration, P.K.; Software, Z.W.; Supervision, P.K.; Validation, R.Z.; Writing—original draft, Z.W.; Writing—review & editing, R.Z. and P.K. All authors have read and agreed to the published version of the manuscript.

**Funding:** This research was funded by the EPSRC, grant EP/P006930/1, "Future Advanced Metrology Hub" and grant EP/K018124/1, "The Light Controlled Factory".

**Acknowledgments:** The authors would like to gratefully acknowledge the industrial collaborators for their contribution as well as the Department of Mechanical Engineering at the University of Bath.

**Conflicts of Interest:** The authors declare no conflicts of interest

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
