# Peer review of "Real-Time Laser Tracker Compensation of Robotic Drilling and Machining"

_jmmp, doi:10.3390/jmmp4030079_

Round 1
Reviewer 1 Report
The paper is good in technical applications but weak in methodologies. There is not much novelty nor significant contributions. There are many experimental applications with nice results.
The main methodology part of this paper in Section 3 and 4 are not written well. The estimation of SMR and TCP offsets is unclear. It seems that these terms are computed straightly forward and are only relative to end-effector flange frame. The key coordinates of the system are the robot root frame, the end-effector flange reference frame and the laser tracker base frame. It is suggested that all coordinate transformations including SMR and TCP offsets are represented by well-defined standard homogeneous transformation, {a}^T_{b}, with specified reference frame {a} and {b}.
Minor: The output of PD controller is not stated.
Author Response
The paper is good in technical applications but weak in methodologies. There is not much novelty nor significant contributions. There are many experimental applications with nice results.
Our response:
We agree that the novelty of the paper was not well explained in the introduction. We will add more detail in the introduction to explain this.
While many papers have been published on machining with industrial robots, very few have attempted to correct their position errors in real-time using external metrology instruments. Of the few papers that have been published, all used more expensive full 6 DOF metrology solutions (Leica trackers or Nikon photogrammetry or 4x modified faro trackers). And even fewer papers presented representative machining results (i.e. aluminium, or other metallic components). Not only is our method lower cost, our results are also very good. Many existing papers also do not describe fully the calibration process of the end-effector, the workflow or the technical details. As an example in this paper [1], representative of typical performance on a similar robot, the authors machined a test piece from a soft material, when 6DOF laser compensation is enabled, only simple slots were milled, and they did not describe how their system is calibrated. Even when machining soft material, they have errors of 0.1mm, where as our results (periphery, figure 27, forces identical to slotting, but more complex path) are below 0.04mm in aluminium. With our paper, we also aim to help other interested parties in creating their own real-time compensation set-ups by sharing our methods, what we learnt, and what accuracy improvements potential problems they can expect.
1. https://doi.org/10.4271/2016-01-2137
The main methodology part of this paper in Section 3 and 4 are not written well. The estimation of SMR and TCP offsets is unclear. It seems that these terms are computed straightly forward and are only relative to end-effector flange frame. The key coordinates of the system are the robot root frame, the end-effector flange reference frame and the laser tracker base frame. It is suggested that all coordinate transformations including SMR and TCP offsets are represented by well-defined standard homogeneous transformation, {a}^T_{b}, with specified reference frame {a} and {b}.
Our response:
It is correct that the SMR and TCP offsets are only relative to the end-effector flange frame. These are the only parameters that need to be computed in our method. We do not need to calibrate any of the six joint parameters or any other parameters of the robot, since we are directly measuring the position error with the laser tracker. We believe the method we developed is both simple and repeatable and should be easy to follow?
Reviewer 2 Report
The paper presents a method for realtime compensation of robot paths for processes such as robotic machining. An experimental setup consists of a Kuka medium/high payload robot KR120 with a reach of 2500 mm, mounted with a SMR and a laser tracker used for the realtime compensation of robot motion.
The paper is technically sound and it provides a lot of details of the experimental setup, the validation setup, and results, as well as the identification of major sources of errors in robot path following applications. The obtained results are very valuable and different types of influential factors on the robot accuracy are explained and experimentally validated in different test setups.
Section 3 – how was it determined that the A5 and A6 are repeatable even more then the laser tracker measurement uncertainty? In Kuka robots these axes are controlled through planetary gearheads or harmonic drives coupled to belt systems. How is the repeatability of this coupled system better that the one for example of A4?
Section 3 – The estimation of the SMR with respect to the robot flange to calculate the exact SMR TCP offset is done trigonometrically with a minimum number of expressions and calculations. If the process of these measurements is done in multiple ways a calculation error could be derived.
Section 4 – is the method developed in your paper scalable to other robot manufacturers? For example is it possible to use this type of realtime compensation on a Fanuc, ABB or Motoman robot concerning their update and control frequencies? Kuka uses RSI which has a high frequency for real time control of 250Hz (Kuka lightweight robots have FRI enabling control frequencies up to 1500 Hz). Furthermore, do you use some realtime operating system in order to reduce the latency of the whole system?
Section 5 - Figure 1 should be annotated as Figure 7.
Section 5 – You state that you use a Win based PC. Windows is known for high latencies and a stochastic nature because the user does not have any control of hundreds of background tasks and processes. What could be the magnitude of the error induced by your non realtime OS?
Section 5 – include in the text the information for the update frequency of the Faro laser tracker.
Line 194 – correct „are be applied“
Section 6.2. – you provide a test with different feed rates. How does the effect of robot payload affect the robot accuracy in the ball bar test?
Section 7.1 – paragraph from line 348 – the anecdotal effect is interesting but it is unclear what exactly happens in this case when the position is “higher up”. This effect should be investigated in more detail and at least elaborated in a sense that you might be able to anticipate the drill holes or positions in other machining operations where such effects appear.
Section 7.1. – paragraph from line 357 – it would be interesting to see the updated results after further calibration of the datum shift.
Section 7.1. – The orientation error is larger with compensation then without compensation. How do you plan on tackling this error in the future? Are you planning on using optical CMM's or other metrology instruments capable of realtime 6D measurements in synergy with the laser tracker to compensate the TCP orientation?
How would your solution scale to a 6D problem with compensation of the orientation, not just the TCP position?
Section 8 – Line 412 – why was the part not optimized for machining? It would be better if you tested your algorithms on parts that do not yield such increased vibration because of their structural and mechanical properties.
Section 9 – you have investigated the effect of robot speed on backlash. You state that future work and the analysis and modelling of robot backlash is required. Have you investigated research dealing with measuring and modelling robot backlash such as: Modeling and compensation of backlash and harmonic drive-induced errors in robotic manipulators - https://doi.org/10.1115/MSEC2014-4123 and Modeling and assessment of the backlash error of an industrial robot https://doi.org/10.1017/S0263574711001287 ? Could you improve your results by implementing existing methods for backlash compensation/modelling?
Author Response
Section 3 – how was it determined that the A5 and A6 are repeatable even more then the laser tracker measurement uncertainty? In Kuka robots these axes are controlled through planetary gearheads or harmonic drives coupled to belt systems. How is the repeatability of this coupled system better that the one for example of A4?
Our response:
We determined this by attaching a SMR to axis A5 and A6, rotating that axis and looking at the best fit circle errors. The errors are typically around 5-10um, better than the measurement uncertainty of the laser tracker. (Note: only the circularity of the axes are important here, not absolute angular accuracy. Therefore, for the purpose of end-effector calibration, with no load, they can be used as references. However, during machining external loads may cause errors on these axes, we rely on the tracker compensation to correct position deviations caused by these errors.
Section 3 – The estimation of the SMR with respect to the robot flange to calculate the exact SMR TCP offset is done trigonometrically with a minimum number of expressions and calculations. If the process of these measurements is done in multiple ways a calculation error could be derived.
Our response:
In the future we would also like to create a full uncertainty budget for the process, using NPL laser tracker model and Monte Carlo simulation.
Section 4 – is the method developed in your paper scalable to other robot manufacturers? For example is it possible to use this type of realtime compensation on a Fanuc, ABB or Motoman robot concerning their update and control frequencies? Kuka uses RSI which has a high frequency for real time control of 250Hz (Kuka lightweight robots have FRI enabling control frequencies up to 1500 Hz). Furthermore, do you use some realtime operating system in order to reduce the latency of the whole system?
Our response:
We believe this is possible for at least ABB and Fanuc, they both have interfaces for compensation. Using a RTOS will improve the performance by reducing jitter, we are currently working towards such a solution. However, since our laser tracker is not synchronised with the robot controller, it is likely we will still have jitter.
Section 5 - Figure 1 should be annotated as Figure 7.
Our response:
Apologies, this will be corrected. The numbering was correct in the submitted version.
Section 5 – You state that you use a Win based PC. Windows is known for high latencies and a stochastic nature because the user does not have any control of hundreds of background tasks and processes. What could be the magnitude of the error induced by your non realtime OS?
Our response:
This can be an issue, but modern PCs are more than fast enough to keep up with the KUKA controller. During very rare occasions when it does happen, the process will just send corrections during the next IPO cycle. This is not a big issue since most corrections are fairly small, <<10um per IPO cycle, so missing one cycle is generally OK. There could be potential issues with RSI, which uses normal Ethernet, and is subject to the nondeterministic Ethernet CSMA/CD process. If there are other users on the same network this can cause further jitters.
A better option will be to use a truly deterministic fieldbus (such as EtherCAT) and a RTOS/PLC), however, as we demonstrate, a normal PC application can be adequate research purposes.
Section 5 – include in the text the information for the update frequency of the Faro laser tracker.
Our response:
Our tracker was set to update at 512Hz, will be added to the paper.
Line 194 – correct „are be applied“
Our response:
Thank you, will be corrected.
Section 6.2. – you provide a test with different feed rates. How does the effect of robot payload affect the robot accuracy in the ball bar test?
Our response:
We have not tested different payloads, since we focused on machining applications, our payload is always the same spindle. This should be investigated in the future.
Section 7.1 – paragraph from line 348 – the anecdotal effect is interesting but it is unclear what exactly happens in this case when the position is “higher up”. This effect should be investigated in more detail and at least elaborated in a sense that you might be able to anticipate the drill holes or positions in other machining operations where such effects appear.
Our response:
We also do not know what exactly is happening, more research into this effect is needed.
Section 7.1. – paragraph from line 357 – it would be interesting to see the updated results after further calibration of the datum shift.
Our response:
We thought about this, but decided to show the errors as they are, and not “massage” the data to much. This represents the errors of the entire process including TCP calibration, part location and CMM datum shift.
Section 7.1. – The orientation error is larger with compensation then without compensation. How do you plan on tackling this error in the future? Are you planning on using optical CMM's or other metrology instruments capable of realtime 6D measurements in synergy with the laser tracker to compensate the TCP orientation? How would your solution scale to a 6D problem with compensation of the orientation, not just the TCP position?
Our response:
Our solution is fully capable of 6D compensation, currently both position and orientation compensation updates are sent to the robot, just with zeros for orientation. As a part of the Future Metrology HUB project, we are exploring potential solutions for 6D correction. The challenge is finding cost effective instruments. Although 6D compensation with Leica 6DOF trackers has been possible for a while, the price tag of £250k is too high for most manufacturers we work with.
Section 8 – Line 412 – why was the part not optimized for machining? It would be better if you tested your algorithms on parts that do not yield such increased vibration because of their structural and mechanical properties.
Our response:
We wanted to produce a part that is representative of typical components used in aerospace. However, with hindsight, the tool path could be more optimized with high speed machining techniques to make the process smoother.
Section 9 – you have investigated the effect of robot speed on backlash. You state that future work and the analysis and modelling of robot backlash is required. Have you investigated research dealing with measuring and modelling robot backlash such as: Modeling and compensation of backlash and harmonic drive-induced errors in robotic manipulators - https://doi.org/10.1115/MSEC2014-4123 and Modeling and assessment of the backlash error of an industrial robot https://doi.org/10.1017/S0263574711001287 ? Could you improve your results by implementing existing methods for backlash compensation/modelling?
Our response:
We have not, but this is something we are actively looking at. The challenge right now is how to implement any compensation process given the limitations of the KUKA controller and having the compensation in a way to not complicate the path programming by the end user. Another interesting paper is this one: https://doi.org/10.1109/EPEPEMC.2016.7752083, which used similar velocity reduction technique as us, but with out laser tracker feedback, therefore their result is not as good, only 10-15% reduction.
Reviewer 3 Report
1. Experimental validation part needs more supplements. line 441-443: “Figure 31 shows the CMM measurement result from with the machining tool path from section overlaid on top, illustrating graphically that the areas of largest errors coincide with sudden changes in toolpath direction.” In order to further verify the conclusion, more tool path strategies such as a spiral is needed to test, in which the sudden change speed resulted from the drastic change of too path direction can be avoided. How many replicates has been produced? How initial workpiece has been machined?
2. Spherically Mounted Reflector (SMR) is a key component to track the errors, but the detailed information for its size and tolerance is lacked (as well as the cutting tool).
3. How does the tracker keep the same pace with continuing motion of the machine tool? Please fully explained in the text.
4. There are many figures which are difficult to understand, please add words in the figure for the items, such as Figure 13, Figure 14, Figure 16, Figure 19, Figure 25, Figure 31.
5. Line 443-444. “A control logic was added to automatically reduce robot speed when a large error is measured, giving the control loop enough time to correct the error, before ramping back to the original speed.” The detailed information for “Automatic control speed” should be illustrated in the text.
6. In literature review and introduction, it is not clear to understand the originality and innovative contribute provided by this paper. What novel thing is revealed? A better discussion is required.
Author Response
1. Experimental validation part needs more supplements. line 441-443: “Figure 31 shows the CMM measurement result from with the machining tool path from section overlaid on top, illustrating graphically that the areas of largest errors coincide with sudden changes in toolpath direction.” In order to further verify the conclusion, more tool path strategies such as a spiral is needed to test, in which the sudden change speed resulted from the drastic change of too path direction can be avoided. How many replicates has been produced? How initial workpiece has been machined?
Our response:
We agree a better toolpath/machining strategy could be used. For the purpose of the paper, we are comparing the performance of the robot with and without compensation, if the toolpaths are the same, the comparison should be valid. We made two pieces, under the same conditions, the only difference is one is without compensation the other one with compensation. From previous experiences and can be seen from the ball-bar results, the robot is very repeatable, therefore we do not expect large variations if more parts are made.
2. Spherically Mounted Reflector (SMR) is a key component to track the errors, but the detailed information for its size and tolerance is lacked (as well as the cutting tool).
Our response:
The SMR error is included in the laser tracker measurement uncertainty, described in section 2. The cutting tool/spindle has a run-out tolerance of 5um, we will include this information the paper.
3. How does the tracker keep the same pace with continuing motion of the machine tool? Please fully explained in the text.
Our response:
We believe the description of the working principles of the laser tracker is not the focus of the paper, we will however add the appropriate references.
4. There are many figures which are difficult to understand, please add words in the figure for the items, such as Figure 13, Figure 14, Figure 16, Figure 19, Figure 25, Figure 31.
Our response:
We will add some labels to the figures where appropriate.
5. Line 443-444. “A control logic was added to automatically reduce robot speed when a large error is measured, giving the control loop enough time to correct the error, before ramping back to the original speed.” The detailed information for “Automatic control speed” should be illustrated in the text.
Our response:
We will add the technical details for automatic speed control in the sections.
6. In literature review and introduction, it is not clear to understand the originality and innovative contribute provided by this paper. What novel thing is revealed? A better discussion is required.
Our response:
We agree that the novelty of the paper was not well explained in the introduction. We will add more detail in the introduction to explain this.
Reviewer 4 Report
The paper shows an experimental setup for using KUKA industrial robot for machining tasks with online feedback from 3DOF laser tracker to improve robot path accuracy. There are some comments about this paper:
1) The paper is similar to another paper for the same author titled:
"Real-time laser tracker compensation of a 3-axis positioning system—dynamic accuracy characterization"
2) The paper doesn't propose any new technique for improving the robot path accuracy over the state-of-the-art techniques. The paper can be considered more as a technical report than a research paper.
3) The references are outdated.
4) The paper doesn't include any comparison with the state-of-the-art research in the same field.
5) The paper has many language errors.
6) Why PD controller is used for processing the feedback data? why not the PID? or only the P or any other adaptive feedback technique, especially the system is claiming online real-time processing.
7) The abstract is mentioning that the main accuracy problem occurs in the aerospace industry. So why the experiment didn't compare performance with other techniques carried out in the aerospace industry on one of the aerospace products?
8) The paper can be considered more as a technical report rather than a research paper because the idea is not novel, it is just the same experiment included in the old papers but with a different robot, even without any performance comparison between old and new tools.
9) There is no comparison between the proposed technique and the state-of-the-art techniques used for the same machining and metrological tasks.
10) The authors should cite their previews papers and add a detailed comparison between the 3 papers.
11) After improving the paper, a new section needs to be added to show the contribution of the paper over the state-of-the-art research.
Author Response
The paper shows an experimental setup for using KUKA industrial robot for machining tasks with online feedback from 3DOF laser tracker to improve robot path accuracy. There are some comments about this paper:
1) The paper is similar to another paper for the same author titled:
"Real-time laser tracker compensation of a 3-axis positioning system—dynamic accuracy characterization"
Our response:
At the Laboratory for Integrated Metrology Applications at the University of Bath, our work focus on improving the accuracy of manufacturing processes through the integration of measurement instruments in manufacturing, so naturally there are some similarities to our work, as they mostly involve measurement, sensors, control, and manufacturing.
We developed the strategy of real-time compensation as a way of reducing defects during machining, rather than inspection post machining which leads to scraps. In the earlier papers, we tested this concept on a small 3 axis machine. This gave us the confidence that the strategy works before applying it to a full-scale robot.
2) The paper doesn't propose any new technique for improving the robot path accuracy over the state-of-the-art techniques. The paper can be considered more as a technical report than a research paper.
Our response:
We agree this was poorly explained in the paper. Although we are not presenting a new concept for improving robot accuracy, there are many novel implementations. First, our new method of path corrections dramatically reduces the effect of jitter due to desync between the robot and the laser tracker. This allows other slower metrology instruments to be used. Secondly, we believe our SMR calibration technique is new, simple, and repeatable. Thirdly, we showed that a cheaper Faro laser tracker without modification can be used compared to all previous work using the more expensive Leica 6DOF systems. This is not a trivial point, especially to our industrial partners, as the cost difference is 5x. Fourthly, our proposed method to mitigate backlash problems are significantly better compared to previous work [1] where only 10-15% reduction was achieved. Lastly, we achieved <50um tolerance on ~90% of an aluminium part made by a medium/high payload, we believe this is a significant achievement, and the results should be shared.
1. https://doi.org/10.1109/EPEPEMC.2016.7752083
3) The references are outdated.
Our response:
Our references date from 2004-2019, if the reviewer would like us to cite specific additional references, please let us know.
4) The paper doesn't include any comparison with the state-of-the-art research in the same field.
Our response:
We agree, we will try to find some examples, maybe indirect comparisons. There are few papers with results that are directly comparable (real-time/online metrology compensation, while machining aluminium/other metal).
5) The paper has many language errors.
Our response:
We will correct any errors found.
6) Why PD controller is used for processing the feedback data? why not the PID? or only the P or any other adaptive feedback technique, especially the system is claiming online real-time processing.
Our response:
The correction vectors sent to the robot are incremental, so the overall effect is a integral loop. We will add better explanation in the paper.
7) The abstract is mentioning that the main accuracy problem occurs in the aerospace industry. So why the experiment didn't compare performance with other techniques carried out in the aerospace industry on one of the aerospace products?
Our response:
Our drilling test was carried out as a comparison with a commercial system on behalf of an aerospace company with a very similar experimental set-up. Our results are significantly better. However, we do not know why the commercial system performed so poorly since we did not conduct the test ourselves. It is unfortunate that we cannot share any information on this or any other trials we carried out with this company. We do agree that comparison of results with other state-of-the-art techniques should be included in the paper.
8) The paper can be considered more as a technical report rather than a research paper because the idea is not novel, it is just the same experiment included in the old papers but with a different robot, even without any performance comparison between old and new tools.
Our response:
We believe the application of a already well defined concept to a new system should on its own be considered "novel", as long as the results produced is valuable to the field of research, even if the results are negative. Although our results are robust and better than state-of-the-art, we did not properly explain this fact. We will correct this.
The reviewer's assertion that the same experiment were repeated is also incorrect. The bulk of the paper is on the drilling and machining performance of an industrial robot, as written in the title of the paper. In the previous papers, no machining tests were carried out, only positioning accuracy tests. In fact, the small machines did not have a machining spindle. Even if we did carry out machining tests in the original papers, it should be apparent that demonstrating machining on a small prototype machine specifically made for experimental purposes will have entirely different results compared to a full-scale robot.
9) There is no comparison between the proposed technique and the state-of-the-art techniques used for the same machining and metrological tasks.
Our response:
See our reply to question 4, as the question is similar.
10) The authors should cite their previews papers and add a detailed comparison between the 3 papers.
Our response:
We will cite our previous papers to provide context, however, the results are not directly comparable as previous explained.
11) After improving the paper, a new section needs to be added to show the contribution of the paper over the state-of-the-art research.
Our response:
We agree that the novelty of the paper was not well explained in the introduction. We will add more detail in the introduction to explain this.
Round 2
Reviewer 1 Report
The revised paper now is fairly well written and the subject is interesting.
The reviewer questions are addressed successfully.
Author Response
Thank you for your comments.
Reviewer 4 Report
The authors did some changes in the paper but there are some comments about the paper that they did not handle:
1) They did not cite or compare some state-of-the-art research in the same field. Some examples of recent research that has not been discussed or cited on the same topic:
"Real Time Pose Control of an Industrial Robotic System for Machining of Large
Scale Components in Aerospace Industry Using Laser Tracker System"
"Robotic Machining: A Review of Recent Progress"
"Compensation for absolute positioning error of industrial robot considering the optimized measurement space"
"ADVANCED SENSING DEVELOPMENT TO SUPPORT ACCURACY ASSESSMENT
FOR INDUSTRIAL ROBOT SYSTEMS "
Author Response
"Real Time Pose Control of an Industrial Robotic System for Machining of Large
Scale Components in Aerospace Industry Using Laser Tracker System"
This paper is already cited
"Robotic Machining: A Review of Recent Progress"
We added this citation in the paper
"Compensation for absolute positioning error of industrial robot considering the optimized measurement space"
This paper is not about machining or real-time compensation, therefore will not be included
"ADVANCED SENSING DEVELOPMENT TO SUPPORT ACCURACY ASSESSMENT
FOR INDUSTRIAL ROBOT SYSTEMS
This paper is published after we submitted our paper. It is about developing a sensor, and does not show any significant results, therefore will not be included.
We have added an additional paper citation "Schneider U, Diaz Posada JR, Drust M, Verl A (2013) Position control of an industrial robot using an optical measurement system for machining purposes. In: International Conference on Manufacturing Research (ICMR), pp. 307-312, Cranfield University, United Kingdom" which is more relevant to our work.